# Blazed oblique plane microscopy reveals scale-invariant inference of brain-wide population activity

Maximilian Hoffmann[1,4,5], Jörg Henninger [1,5], Johannes Veith [1,2], Lars Richter [3] & Benjamin Judkewitz [1]✉

Due to the size and opacity of vertebrate brains, it has until now been impossible to simultaneously record neuronal activity at cellular resolution across the entire adult brain. As a result, scientists are forced to choose between cellular-resolution microscopy over limited fields-of-view or whole-brain imaging at coarse-grained resolution. Bridging the gap between these spatial scales of understanding remains a major challenge in neuroscience. Here, we introduce blazed oblique plane microscopy to perform brain-wide recording of neuronal activity at cellular resolution in an adult vertebrate. Contrary to common belief, we find that inferences of neuronal population activity are near-independent of spatial scale: a set of randomly sampled neurons has a comparable predictive power as the same number of coarse-grained macrovoxels. Our work thus links cellular resolution with brain-wide scope, challenges the prevailing view that macroscale methods are generally inferior to microscale techniques and underscores the value of multiscale approaches to studying brain-wide activity.

Our current knowledge about brain function covers different spatial scales, ranging from the microscopic to the macroscopic level. On the macroscopic level, advances in human brain imaging, such as fMRI, have increased our knowledge about the roles of different brain regions, their functional networks and global activity patterns[1–4]. Despite the important insights that are being gained from such macroscopic measurements, they can only provide a coarse-grained picture of a brain that is made of individual cells. For example, an fMRI voxel contains thousands of distinct cells, whose activity is averaged into one value[5]. In contrast, microscopic studies complement this global picture by studying local brain circuits with single-cell resolution. Multi-electrode recordings and two-photon calcium imaging[6–9] have uncovered a high diversity of functional properties even among nearby cells and within local microcircuits. However, the field of view (FOV) and penetration depth limit two-photon microscopy to only a small part of the mammalian brain volume and even the latest multi-electrode probes with thousands of channels can only record far less than a percentage of all the cells in the mammalian brain ($7 \times 10^7$ in the mouse). As a result, both macroscopic as well as microscopic recordings represent a severe spatial subsampling of total brain activity, be it by local averaging or by extreme selection.

How much can we learn about the dynamics of the whole system if we only record a small fraction? If we are forced to subsample, what type of subsampling should we choose? Given a limited number of recording channels, are single cells or large voxels more predictive of brain-wide cellular activity? Answering these questions requires combining microscopic single-cell resolution with a macroscopic brain-wide range. This, until now, has only been possible in invertebrates or developing zebrafish larvae[10–13], but has not been achieved in any adult vertebrate.

[1]Einstein Center for Neurosciences, NeuroCure Cluster of Excellence, Charité – Universitätsmedizin Berlin, Berlin, Germany. [2]Department of Biology, Humboldt University Berlin, Berlin, Germany. [3]Department of Chemistry and Center for NanoScience, Ludwig Maximilians University, Munich, Germany. [4]Present address: Rockefeller University, New York, USA. [5]These authors contributed equally: Maximilian Hoffmann, Jörg Henninger. ✉e-mail: benjamin.judkewitz@charite.de

Addressing this limitation, we and others recently introduced the teleost *Danionella cerebrum* (DC) as a new model for neuroscience[14–18]. DC has the smallest known adult vertebrate brain which, at ~2 mm length, is only twice as long as that of a 5-day-old zebrafish larva. The brain is transparent and optically accessible from the top, as it is not covered by a dorsal skull.

However, light-sheet microscopy (LSM)[19], the established method for volumetric imaging in zebrafish larvae, requires samples to be transparent from multiple sides. In DC, the excitation light would have to be sent through the scattering lateral skull, not the dorsal natural window. As a result, there is no suitable high-speed volumetric recording technique that can cover a major part of the adult brain.

Here, we develop blazed oblique plane microscopy, which overcomes these limitations and enables us to perform the first brain-wide cellular activity measurements in an adult vertebrate at cellular resolution. Using this technique, we measure spontaneous activity across the brain of adult DC and investigate the relationship between cellular activity and macroscopic phenomena. We artificially subsample and voxelize the data to ask whether microscopic or macroscopic features are more informative of brain-wide population activity.

## Results

### Blazed oblique plane microscopy

In order to record cellular activity with calcium sensors throughout the brain of adult DC we need to sample a volume of approximately $2.2 \times 1.2 \times 0.65$ mm$^3$ at ≥ 1 Hz.

Raster scanning microscopy, such as two photon imaging[6,9,20], is one of the most widespread techniques for fluorescence imaging of microscopic brain activity, but its speed and signal level are limited by the need for point-scanning, by the fluorescence lifetime and by thermal damage thresholds of the tissue. Light sheet microscopy (LSM)[19], overcomes this limitation by capturing entire planes of the specimen at once using a camera, but the need for an orthogonal excitation arm limits its use to small samples that are transparent from multiple sides—excluding nearly all vertebrates.

Oblique plane microscopy (OPM), a variant of LSM, circumvents these constraints by exciting and imaging oblique planes in the specimen through the same objective[21–30] (Fig. 1a). Because the resulting image plane is also oblique, it cannot be imaged onto a camera sensor using conventional approaches. OPMs therefore employ a remote refocusing step (Fig. 1b) in which an intermediate image of the oblique plane is created using a secondary microscope objective (Obj2). This intermediate image plane is aligned into the focal plane of a tertiary detection objective (Obj3), imaging it onto a camera sensor.

The magnification between the specimen and the intermediate image plane is typically chosen to fulfill the condition of unit angular magnification[31] ($M = n_1/n_2$, the ratio of refractive indices at the sample and intermediate image plane, respectively). This ensures that first-order spherical aberrations are canceled[32] and even the points of the oblique plane that are outside of the native focal plane are imaged accurately. The refocusing step, however, leads to a loss of light: Some light will necessarily propagate outside of the acceptance cone of Obj3 (Fig. 1b). This implies that this re-imaging solution cannot be used below an NA of 0.5·n for Obj1 and Obj2, because the loss would become total (Fig. 1c, green curve, Supplementary Fig. 1a). At the same time, due to the constraints of optical design, the achievable FOV is inversely proportional to the NA of an objective (Fig. 1c, blue dots). This relationship limits the achievable FOV of conventional OPM to about 1 mm (Fig. 1c)[33,34].

Multiple strategies have recently been proposed to mitigate this trade-off. Objectives with a high index of refraction immersion medium at Obj3 can increase the light efficiency by refracting the light cone, but their FOV is still limited to <1 mm[27,35,36]. Alternatively, the sample can be de-magnified onto Obj2, but the deviation from unit angular magnification rules out aberration-free imaging[37]. A recent approach using reflective diffraction gratings is limited to only operating at low NAs (NA < 0.3)[38]. As a result, none of the existing approaches are suitable for imaging the entire DC brain at cellular resolution.

We reasoned that, to efficiently reimage the oblique image plane with minimal loss, we need to tilt the illumination cone without affecting the image intensity distribution. In terms of wave optics, this

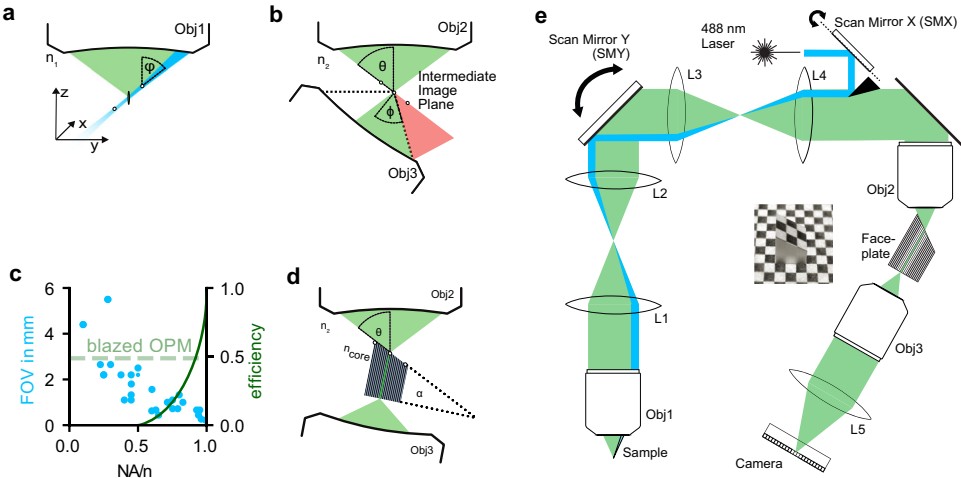

**Fig. 1 | Blazed oblique plane microscopy. a** Fluorescence is excited by a light sheet (blue) focused through the primary microscope objective (Obj1) at an angle φ. The imaged plane (white circles) in the medium of refractive index $n_1$ is therefore oblique. Emitted fluorescence (green) is then collected through the same lens. **b** To image the oblique plane onto a camera, an intermediate image (white circles) is created by a secondary objective (Obj2) at an angle θ in a medium with refractive index $n_2$. This plane is then brought to lie in the image plane of a tertiary microscope objective with an acceptance angle φ. This leads to loss of light (red) for all φ <90°. **c** The overall efficiency of the re-imaging strategy is critically dependent on the NA of Obj2 and Obj3 (green curve). At the same time, the FOV scales inversely with the objective NA, shown in blue for selected, commonly used microscope objectives. **d** Blazed OPM employs a custom fiber optic faceplate (FP) with a core refractive index $n_{core}$, that is cut at an angle α. α is chosen to minimize the coupling losses into the array of multi-mode fibers. The angled facet of FP is positioned at the intermediate image plane after Obj2. The intensity distribution at the intermediate image plane is then transmitted to the other end of the FP, where it is imaged by Obj3. **e** Complete setup consisting of the microscope objective lenses (Obj1-3), relay lenses (L1-L4, $f = 200$ mm), scan mirrors (SMX/SMY), the excitation laser, the fiber-optic faceplate, and a high-speed camera. Inset: photograph of the FP placed on a printout of a checkerboard pattern, illustrating its image transfer capability.

implies applying a strong phase ramp (also "blazed" phase) to the intermediate image, without changing its amplitude. Blazed OPM achieves this goal via a specially fabricated (but easily mass-producible) fiber optical faceplate (Fig. 1d, e). This faceplate (FP) is a rigid array of small optical fibers with a diameter of 2.5 µm (Fig. 1e inset). It can transfer light intensity distributions from one plane to another without any additional optical system, while applying a phase ramp from the input to the output plane. If the intermediate image plane of a blazed OPM is brought to lie coplanar to the oblique surface of the FP, single fluorescence foci are coupled into the individual fibers of the FP. The opposite end of the FP can then be imaged onto a camera by the collection imaging system consisting of an objective (Obj3) and a tube lens (L5).

The geometry of the FP needs to be optimized to ensure that most of the light is propagating within the coupling angle. During coupling, the light is refracted at the boundary between the surrounding medium (air, $n_2 = 1$) and the fiber core ($n_{core} = 1.81$). Assuming that the mean incidence angle of the incoming cone is the acceptance angle θ, selecting the face angle α to be $\alpha = \arcsin(\cos(\theta) \cdot n_2/n_{core})$ ensures that the light is optimally coupled into the faceplate (Supplementary Fig. 1b). Under these conditions, we measured its light efficiency to be at 48% (See Methods).

Blazed oblique plane microscopy can be combined with different primary objectives (Obj1, Supplementary Fig. 2), which are close to the required unit angular magnification that ensures the cancellation of first-order spherical aberrations, but here we use a 16×/0.8 NA objective for optimal trade-off between FOV and axial resolution. To assess the performance of this configuration, we imaged a sample of fluorescent beads (∅ = 1 µm) (Supplementary Fig. 2) and determined a resolution of $2.8 \pm 1 \times 2.4 \pm 0.9 \times 13.2 \pm 2.8\ \mu m^3$ (FWHM, $n = 17684$ beads) across a FOV of $2.1 \times 1.7 \times 0.8\ mm^3$, along x, y, and z, respectively. This resolution is consistent with the limit imposed by the fiber pitch (in this case: 2.5 µm, demagnified onto the sample by factor 1.6). Optical sectioning is important for volumetric imaging of densely labeled objects. We quantified the sectioning capability in Supplementary Fig. 3.

Using this approach, we imaged the brain of adult DC at a volume rate of 1 Hz. With transgenic animals expressing a nuclear-localized calcium indicator (elavl3:H2B-GCaMP6s × tyr$^{-/-}$), we recorded spontaneous activity from up to 41 k active neurons throughout distant brain regions up to a depth of ~400–500 µm, including the brainstem, tectum, pallium, and diencephalon, excluding the ventral hypothalamus (Fig. 2a–d, Supplementary Figs. 4–7 and Supplementary Movie 1). Blazed OPM thus enabled the first brain-wide recording of cellular activity in an adult vertebrate.

## Spatial scale of correlations

To understand how the activity of sparsely sampled cells or coarse-grained voxels relates to brain-wide population activity, we started by characterizing functional coupling across spatial scales. In macroscopic studies, such as fMRI or wide field calcium imaging, functional coupling (FC) between pairs of brain regions or voxels is commonly quantified by pairwise Pearson correlations, also referred to as "functional connectivity".

To investigate such correlations across scales, including cellular resolution, we recorded calcium signals from ~150 k neurons in 6 animals (17 k – 41 k spontaneously active neurons per animal, of an estimated total 650 k neurons[14]). Analyzing FC across ~10$^9$ pairs of individual cells, we found that its distance-dependent decay can be fit by a power law (a line on a log–log plot), ranging from microscopic scale to macroscopic, brain-wide distances (Fig. 3a). This is in line with macroscopic fMRI studies in humans that reported scale-free correlation decay at a millimeter resolution[39], and demonstrates that the power-law relationship extends from the global level down to the microscopic level[40] in a mature circuit. To visualize the scale and spatial granularity of correlations, we created maps of brain-wide FC values for individual cells (shown in Fig. 3b–d for three example cells with local and long-range coupling).

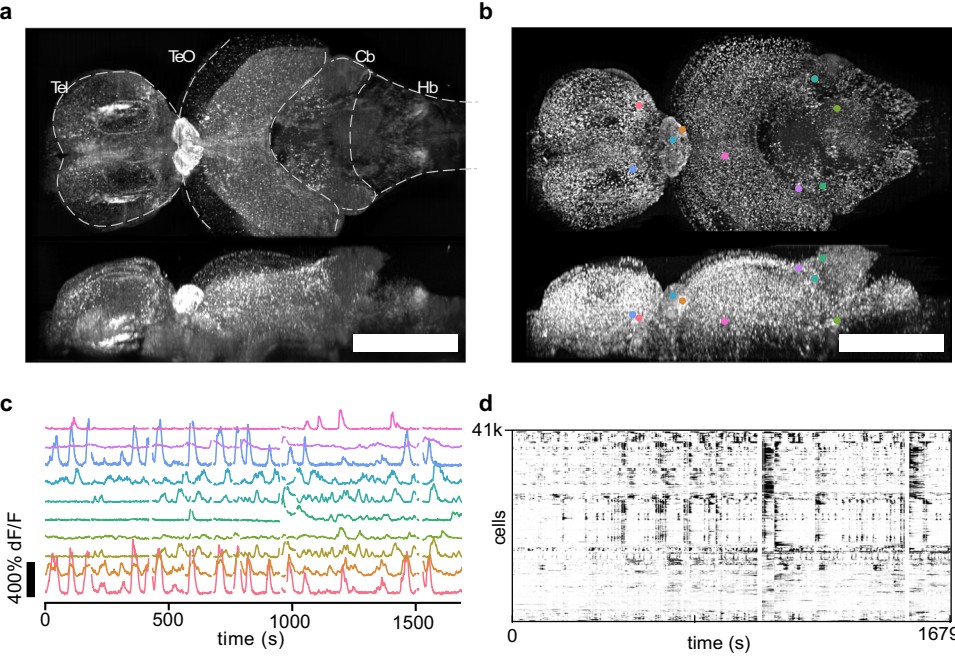

**Fig. 2 | Brain wide imaging of *Danionella cerebrum* (DC). a** Orthoprojections (maximum intensity) of the time series mean of an adult DC (line elavl3:H2B-GCaMP6s x tyr-). The recording lasted 38 min at a volume rate of 1 Hz. The FOV encompassed major brain regions, such as the telencephalon (Tel), optic tectum (TeO), cerebellum (Cb) and hindbrain (HB). **b** Local temporal correlation map serving as the segmentation map of the brain. **c** Fluorescence dynamics from ten neurons at different locations marked in (b). **d** We segmented ~ 40 k putative neurons and extracted their time dependent fluorescence (centered dF/F, displayed between 0 and 30%; cells sorted by 1D embedding; white bands: excluded frames due to motion artifacts). Scale bar: 500 µm.

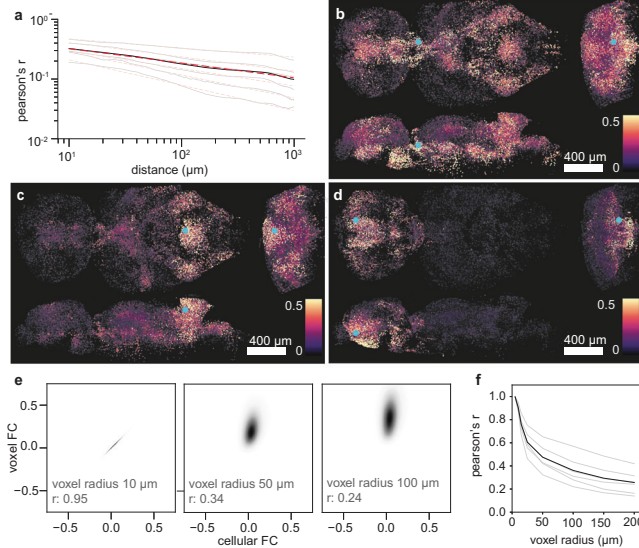

**Fig. 3 | Spatial scale of correlations: the relationship between microscale and macroscale functional coupling. a** Average pairwise Pearson's correlation as a function of distance, for 6 animals and their mean, power law fits dashed in red (exponent $0.28 \pm 0.09$ (mean $\pm$ std). **b–d** Maps of brain-wide FC values for three example cells with local and long-range coupling (cell location marked by cyan dot). **e** Scatter across $\sim 10^9$ FC values (normalized density), where each point represents a pair of cellular and corresponding voxel FC values—for three different neighborhood radii. For small voxel radii, the local neighborhood activity trivially corresponds to single cell traces (left). As the neighborhood size increases, FC values increase and deviate from their cell-scale counterparts. **f** Secondary Pearson correlation between the cell and voxel FC values (the correlation of the scatter in **e**) as a function of neighborhood size for all animals in **a** overlayed with the mean decay. Source data are provided as a Source Data file.

We then sought to understand the link between microscopic FC at cellular resolution and macroscopic FC values one would obtain from larger voxels. We created a synthetic dataset by averaging traces of nearby cells within a given radius. This dataset has the same number of traces as the original data, yet each trace no longer represents a single cell, but the average activity of all cells within a given radius from the single cell. This manipulation allowed us to compare FC values between pairs of cells to FC values between pairs of corresponding voxels that are centered on the same locations. Density plots comparing $\sim 10^9$ microscopic FC values with their corresponding macroscopic FC values are shown in Fig. 3e. For small voxel radii (<10 µm), microscopic and "macroscopic" FC values are identical, because voxels contain only one cell. However, once the voxel size increases beyond cell scale, macroscopic FC values begin to increase, and deviate from their microscopic counterparts. Thus, the secondary correlation between microscopic and macroscopic FC values rapidly drops as the voxel size increases (Fig. 3f).

The observation that spatial averages only poorly represent cellular-level activity could lead to an interpretation that macroscale measurements "destroy" information about the microscale. However, the fact that there is no direct correspondence between macroscale and microscale activity need not imply that voxels cannot be used to make inferences about cellular-level brain activity. For example, the activity of cells that are functionally coupled to multiple other cells across multiple brain regions might still be well predicted by spatially averaged brain-region activity traces.

## Scale-invariant predictions of brain activity

We therefore asked how well cellular activity can be predicted from voxels and how different scales of spatial granularity—from single cells to large voxels—compare in terms of their predictive power. We

started by quantifying the dimensionality of our dataset using bi-cross-validated principal component regression[41,42] to predict the activity of all recorded cells. This analysis revealed a lower bound of $400 \pm 100$ (mean + std, $n = 6$ animals) predictive dimensions, explaining over 30% of the variance of the brain-wide spontaneous activity in the held-out dataset. To understand how well brain activity can be predicted from limited amounts of data at varying spatial scale, we then subsampled and discretized our data (Fig. 4a), and tested predictive power with linear ridge regression. Our performance measure was the fraction of variance explained when predicting global brain activity at single-cell resolution (i.e. the variance-weighted average $R^2$ of all recorded cells). First, we used sets of randomly sampled cells as predictors. As expected, the predictive power increased as a function of predictor cell number (Fig. 4c). We then discretized our predictor dataset into non-overlapping macrovoxels, while keeping the same prediction target: all cellular-resolution activity that was recorded across the brain. In the resampled predictor dataset, the activity of all cells assigned to a given voxel was averaged into one-time trace. We observed that prediction power decreased when the voxel size increased from cellular scale ($\sim 5$–10 µm) to hundreds of µm (Fig. 4b). This effect could have different reasons: loss of microscale information, or the reduction of predictor count (the brain fits more small voxels than big voxels). To distinguish these two, we plotted the data as a function of voxel number and observed that the variance explained by a given voxel number was close to the variance explained by the same number of randomly selected single cells (Fig. 4c; cell/voxel $R^2$-ratio of $1.0 \pm 0.1$, $n = 6$ animals). In this analysis, voxel size and voxel number are still linked because a given voxel size will parcellate the brain into a set number of voxels. To decouple both quantities, we decided to compare random samples of non-overlapping voxels at varying sizes (Fig. 4d). Remarkably, we found that the $R^2$ remains near-constant for a given number of predictor channels, irrespective of spatial granularity. For example, 500 randomly sampled cells have a comparable predictive power as 500 macrovoxels of 100 µm side length (Fig. 4d, Fig. 4f). We observed this effect across a wide range of tested predictor numbers, with higher numbers limiting the size of non-overlapping macrovoxels.

How can these results be explained? An intuitively appealing hypothesis is that the predictive power of voxels is related to the spatial structure of correlations: as we show above (Fig. 3a), nearby cells are highly correlated and the decay of average correlations over distance can be approximated by a power law function. The scale-invariant predictions could be a corollary of this spatial organization. Scrambling of spatial structure should then abolish our ability to predict brain activity with voxel averages.

The alternative hypothesis is that our results are independent of the spatial structure of brain activity. In this view, voxel traces, which are averages of local assemblies, should be as informative as averages of spatially random assemblies. A sampling of k cells, k voxels, or k random assembly averages can all be interpreted as a form of dimensionality reduction or projection into a k-dimensional space. Recent work[43,44] argued that neuronal subsampling can be modeled as *random projection*, a well-studied dimensionality reduction technique[45–48], under the simplifying assumption that high-dimensional brain activity is randomly oriented with respect to the projection axes. To the extent that this model holds for the different types of subsampling, they might all provide equivalent results in the prediction of brain-wide activity.

To distinguish between these two hypotheses, we repeated the coarse-graining analysis of Fig. 4d for a spatially shuffled dataset, in which we randomized the assignment between cell traces and cell coordinates. In this analysis, a "voxel" trace no longer corresponds to the average activity of a local cell assembly, but to the average activity of a spatially random assembly containing the same number of neurons. We found that spatially random assemblies reach a comparable

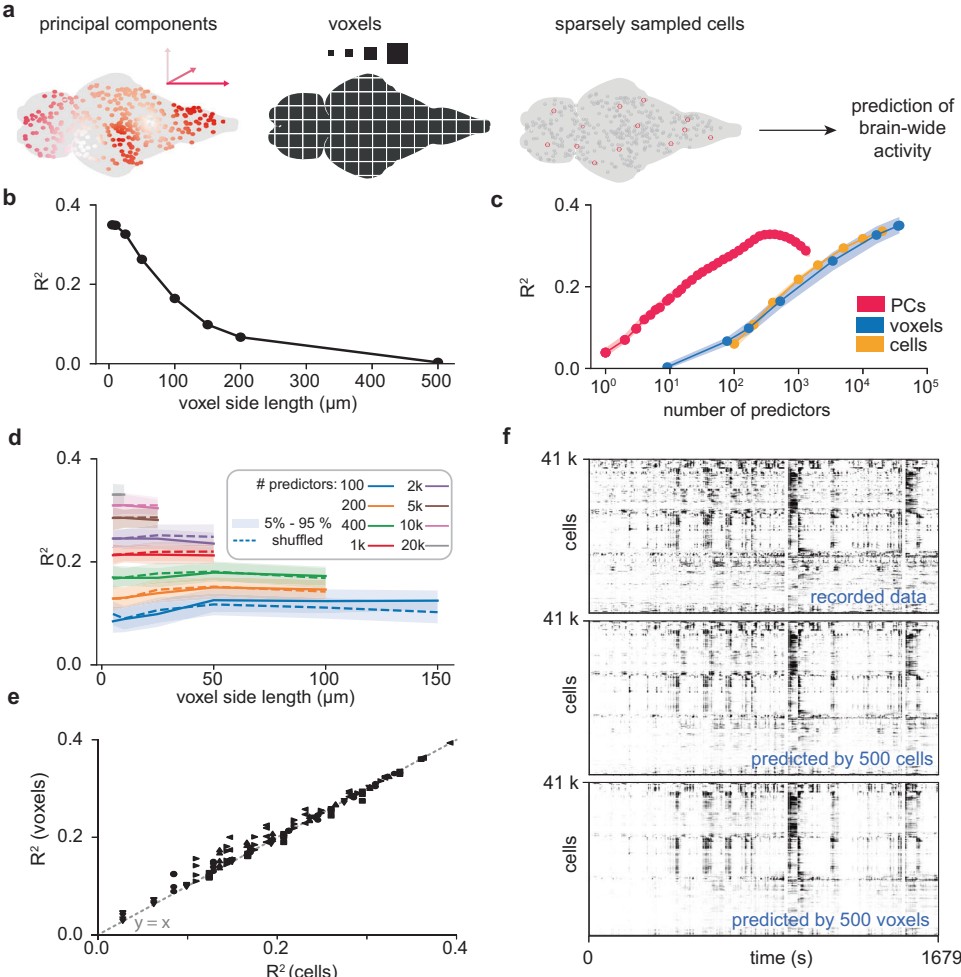

**Fig. 4 | Scale-invariant predictions: inferring brain-wide activity after sub-sampling and coarse-graining. a** Schematic overview of subsampling approach. Our aim was to predict all recorded cellular activity via cross-validated regression, quantified by the fraction of brain-wide variance explained ($R^2$). Tested predictors included principal components, voxels of varying sizes, and sparsely sampled cells. **b** Variance explained as a function of voxel size. **c** Variance explained as a function of predictor number (principal components, non-empty voxels, randomly sampled cells). Data are presented as mean and the range between 5th and 95th percentile. **d** Variance explained as a function of voxel size, from μm-range (individual cells) to 150 μm large macrovoxels. Different colors indicate different numbers of randomly sampled voxels of a given size. Additionally, we computed the same metric after randomly shuffling the correspondence between temporal activity and spatial coordinate, thereby destroying spatial structure (dashed line). Data are presented as mean and the range between 5th and 95th percentile. **e** Correlation plot for the $R^2$ values achieved with the same count of either cells or voxels of different size. **f** Activity maps of recorded activity (top), and predicted activity using 500 cells and 500 macrovoxels of 100 μm size (middle and bottom). Source data are provided as a Source Data file.

$R^2$ as same-size voxels ($R^2$-ratio $1.0 \pm 0.1$; Supplementary Fig. 9; Fig. 4d, dashed lines).

In summary, predictive power is maximized by maximizing predictor number, highlighting the benefit of brain-wide recordings at cellular resolution. However, in a scenario where the number of predictors is limited, as in the vast majority of neuronal recordings in vertebrates, we observed that randomly sampled cells are nearly as predictive of brain-wide activity as macrovoxels. This effect was not abolished by spatial shuffling of brain activity.

## Scale-invariant distortion of neuronal population geometry

Time-varying neuronal population activity can be viewed as a trajectory of the population activity vector through a high-dimensional state space, whose geometry is the subject of an active area of research[8,49–56]. It is therefore natural to ask whether the scale-invariance we observed during prediction also applies to measures of neuronal population geometry.

Whenever the number of recorded channels is smaller than the number of neurons in the brain, a measurement can be seen as a low-dimensional embedding of the high-dimensional full brain activity (Fig. 5a). Conceptualizing subsampling and coarse-graining as forms of dimensionality reduction suggests that they could be compared by the commonly used distortion measure[45,57], which quantifies how the pairwise Euclidean distances in the original space are distorted in the embedded space (see Methods). We measured the distortion for subsampled and coarse-grained data and found that the distortion for a given predictor count was comparable between cells and voxels. For a given predictor count, the ratio of cell to voxel distortion ($1.2 \pm 0.2$, mean ± std.) was close to unity (Fig. 5b, c).

As an alternative measure, we quantified how well subsampling or coarse-graining preserves the correlations between pairs of time points, an approach that is analogous to representational similarity analysis[58]. We calculated the time-by-time correlation matrices for subsampled data and quantified their similarity to the original (non-subsampled data) correlation matrix. Here, too, we found that the similarity ratio between cells and voxels ($0.9 \pm 0.1$, mean ± std.) was close to unity.

Thus, consistent with our results on prediction, sparse sampling was comparable to coarse-graining in preserving the geometry of high-dimensional brain activity.

## Discussion

It is a widely held[59,60], but not uncontested[61] belief that the totality of neuronal activity represents one of the most important levels of description for understanding brain function. Yet in the vast majority of vertebrate models it is currently not possible to record even a percentage of all neurons. Recording a large part of rodent or primate neurons will likely stay impossible for the foreseeable future. Thus, any recording is an implicit subsampling of the total activity, be it by local averaging (e.g. mesoscopic calcium imaging, fMRI, functional ultrasound), local sampling (e.g. multi-photon microscopy), or spatially distributed sparse sampling (e.g. electrode arrays)[6,8,62–66].

How much of the total information does subsampling capture and do the different subsampling methods differ? We found that a given number of randomly sampled neurons provides similar information for inferring brain-wide activity as measuring the same number of macrovoxels. This finding has practical implications for scientists working on other vertebrates, who are not able to record brain-wide cellular activity, but have to make choices between microscopic and macroscopic methods. Beyond practical implications for scientists, subsampling also poses an implicit challenge for neuronal wiring. Like scientists, neurons and local assemblies cannot directly sample from the entire brain, but receive inputs from a sparse selection of pre-synaptic cells via local and long-range connections[43].

Our study challenges the common notion that macroscale approaches are generally inferior to microscale techniques, but it does not suggest that they are equivalent. Voxels will never be able to compete with cellular-resolution measurements when it comes to characterizing the function of individual cells. However, a growing number of studies[8,49–56] investigate brain activity in the framework of neuronal population geometry and activity manifolds, which no longer refer to individual cells, but to their collective high-dimensional activity and its evolution over time. In this framework, our results

suggest that the number of randomly sampled channels matters more than their spatial resolution.

In small animals, the number of recorded channels is maximized by high-resolution brain-wide microscopy. In the majority of vertebrates, however, non-trivial trade-offs have to be made between resolution, field-of-view, and the number of parallel recordings. In this case, the highest resolution may not always provide the best estimate of neuronal population geometry.

To achieve brain-wide imaging in DC, we developed blazed OPM, which reaches multi-millimeter FOVs at cellular resolution. In contrast to conventional OPM[21], whose efficiency approaches zero as the NA/n decreases towards 0.5, our approach allows us to refocus the oblique plane at all NAs. We therefore are able to choose primary objectives with intermediate NA and achieve imaging volumes up to 4 mm³ (in clear media), larger than in previous high-NA OPMs[23,26,27,29].

These advantages enabled the first volumetric recording of brain-wide neuronal activity in an adult vertebrate, opening up a range of possible studies that link microscopic and macroscopic scales[67,68]. In this study, blazed OPM allowed us to investigate the effect of spatial granularity on our ability to infer brain-wide activity. Because prediction involves knowledge of data covariance, this question could not be answered by tiling sequential recordings over limited fields-of-view, which would lack simultaneity. Nor could it be answered by global recordings at a coarse-grained level because they lack cellular resolution.

Calcium imaging comes with common caveats that should be considered when interpreting our results. As with other linear microscopy techniques, the depth penetration of OPM is limited by optical scattering, which prevents us from recording cells in dense nuclei and ventral most areas of the brain below 400–500 μm, such as the ventral hypothalamus (see Supplementary Figs. 4-7). The temporal resolution of our recordings is limited by the dynamics of the sensor (~4 s decay time for the sensor used here), limiting us to statements about slowly varying activity. With faster sensors for calcium[69], neurotransmitters[70,71], or voltage[72], blazed OPM could be operated at higher speeds, constrained mainly by the desired exposure time and camera frame rate. Future studies might then be

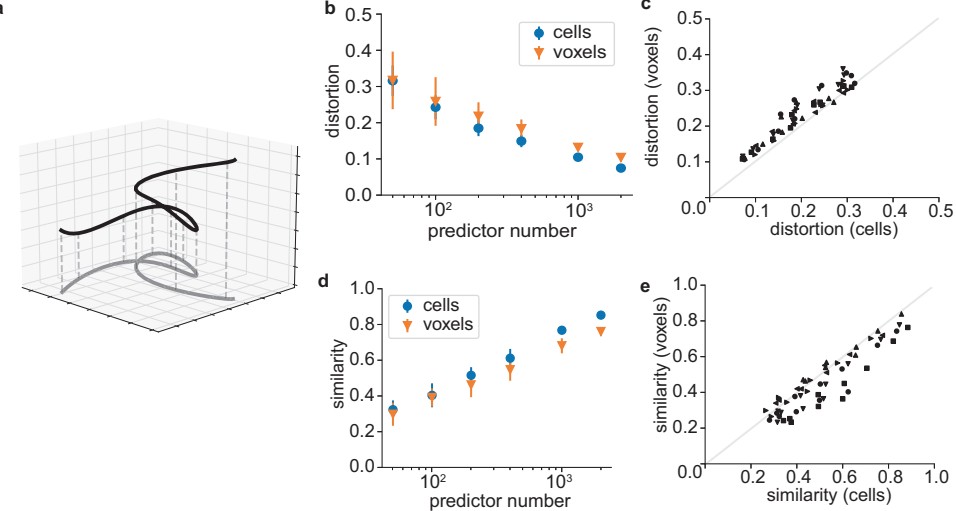

**Fig. 5 | Scale-invariant distortion of neuronal population geometry. a** Schematic of dimensionality reduction (here: from 3D to 2D), which can distort the geometry of the original data. **b** Distortion as a function of predictor number, for subsampled cells or subsampled 50 μm voxels. **c** Comparison between cell and voxel distortion across predictor numbers in b, voxel sizes (50 μm, 100 μm, 150 μm) and animals.

**d** Similarity as a function of predictor number, for subsampled cells or subsampled 50 μm voxels. **e** Comparison between cell and voxel similarity across predictor numbers in d, voxel sizes (50 μm, 100 μm, 150 μm) and animals. Source data are provided as a Source Data file.

able to extend our analysis to the subsecond range and also study the interdependence between spatial and temporal resolution when predicting brain-wide activity.

# Methods

## Ethical Statement
All animal experiments conformed to Berlin state, German federal, and European Union animal welfare regulations and were approved by the LAGeSo, the Berlin authority for animal experiments.

## Optical setup
Our design consists of three 4 f imaging systems. As the primary microscope objective (Obj1) facing the specimen, we employ either a 10x (f = 10 mm, NA = 0.5, water-immersion, CFI Plan Apochromat 10XC Glyc) or a 16× (f = 12.5 mm, NA = 0.8, water-immersion, 16X CFI LWD Plan Fluorite, Nikon) objective lens. The excitation laser (06-MLD, 488 nm, Cobolt) hits a scanning mirror (SMX, 6 mm, 8315 K, Cambridge Technology) and is subsequently reflected into the microscope via a pick-off mirror (PM). The center of SMX is imaged onto the back-focal aperture (BFP) of Obj1 · via two 4-f systems (L1-L2, L3-L4, all TTL200MP, Thorlabs Inc., f = 200 mm). SMX is positioned off-axis in order to create the oblique plane. A fast-scanning SMX, therefore, creates an oblique light sheet within the specimen, exciting fluorescence. The laser beam forming the light sheet had a waist of $\omega_{10x}$ = 7.9 µm and $\omega_{16x}$ = 7.3 µm (excitation NA ~ 0.033) resulting in a calculated Rayleigh range of 400 µm and 350 µm respectively. The angle of the oblique plane in the specimen was $\varphi_{10x}$ = 21° and $\varphi_{16x}$ = 33°. This oblique plane is then imaged onto the intermediate imaging plane of Obj2 (CFI Plan Apo Lambda 10X, f = 10 mm, NA = 0.45) at an angle of θ = 27°.

If the intermediate image plane of an OPM is brought to lie coplanar to the oblique surface of the faceplate (FP, custom-made, material: 24AS, Schott AG, $n_{core}$ = 1.81, $n_{cladding}$ = 1.48, NA 1.0, diameter 2.5 µm, core-cladding ratio 0.7, cut angle $\alpha$ = 35°), single fluorescence foci are coupled into the individual fibers of the FP. The FP material was chosen as the one with the smallest available glass fiber pitch. The opposite end of the FP can then be imaged onto a camera (CB262CG-GP-X8G3, Ximea, pixel pitch = 2.5 µm, 5120 ×5120 pixel) by the collection imaging system consisting of an objective (Obj3, CFI Plan Apo Lambda 10×, f = 20 mm, NA = 0.45) and a tube lens (L5, XLFLUOR 4X, f = 45 mm, NA = 0.28, Olympus). The effective magnification of 2.25 guarantees that each fiber is imaged at the Nyquist criterion.

Volumetric imaging is enabled by a second galvanometric mirror (SMY, 25 mm diameter, 6240H, Cambridge Technology), which is conjugated to the BFP of the primary objective and allows to scan the light sheet throughout the specimen and to de-scan the emission light onto the static intermediate image plane. The synchronization of scanning and camera image acquisition are controlled via custom software (Python) and a data acquisition card (NI USB-6363, National Instruments).

All camera images are recorded with an exposure time of 3 ms with an additional read-out time of 0.145 ms. Our camera has a 10-bit range at the read-out layer. The acquired data is mapped onto a 8-bit range before being transferred to the host computer using a custom linear LUT that fixes the gain to around 5e⁻/count.

To reach a desired optical resolution, fiber diameter and pixel size need to be chosen accordingly, considering the Nyquist sampling limit. The fiber pitch should be chosen to fulfill p ≤ M·r/2, where p is the fiber pitch, M is the magnification from the sample plane to the faceplate surface and r is the size of the smallest object to be resolved. Similarly, the camera pixel size and the magnification of the tertiary imaging system should be chosen such that the magnified fiber pitch on the camera surface corresponds to ~2x the pixel pitch or more.

## Characterisation of image transfer efficiency
In order to characterize the efficiency of FP-based image transfer, we used a fiber-coupled green laser (532 nm, MGL-DS-532, CNI, coupled via P3-405BFC, Thorlabs). The output of the fiber was then collimated using a microscope objective (2X, 0.1 NA, 56.3 mm WD, Thorlabs), with the same 20 mm back focal aperture diameter as objective Obj2. This collimated output was then coupled into the system at the back focal aperture of Obj1. The intensity of the resulting laser focus at the intermediate imaging plane was measured using our tertiary imaging system without the FP (coaxially aligned with the incoming beam). We then inserted the FP and measured the intensity of the light as it would be imaged with blazed OPM. The tertiary imaging system was positioned such that the laser was spread out over multiple fiber cores at the front of the FP to ensure that the measured efficiency accounts for the fill fraction and related losses.

## Characterisation of resolution
Fluorescent beads (diameter 1 µm) were dispersed in a poly-acrylamide gel between a glass slide and a coverslip, separated by a silicon spacer. A stack of the whole accessible image volume was taken. The stack was then processed as described below, but not deconvolved. The stack was thresholded at the 99.99th percentile and all connected components were segmented out within a ROI of 33 × 33 × 303 µm³ (X × Y × Z) For the quantification of the lateral resolution each ROI was maximum intensity projected along z. The lateral resolution in x and y were determined as the full width at half maximum (FWHM) of the Gaussian fit through the line plot along the maximum of this projection. For each bead the axial sectioning capability was determined as the FWHM of the Gaussian fit of the sum of all pixel values along the XY planes. Beads were randomly distributed in space, and with respect to fiber facets.

## Post-processing
After the data is acquired we execute several post-processing steps. For every acquired frame, the camera background is subtracted and multiplicative FP artefacts are corrected. FP artifacts stem from (a) a mild Moire pattern due to the alignment between fiber faces and camera pixels and (b) dust or surface defects. To correct for these artifacts, a reference grid image is obtained by imaging a homogeneously fluorescent object or homogeneously illuminating the input face of the FP with a green LED and recording a correction pattern on the camera. The image is normalized to a maximum value of 1 and clipped to contain values ≥ 0.5 (to avoid dividing by small values, where dark pixels in the middle of a surface defect likely contain no information). Each camera frame is then corrected by dividing the background-subtracted frame by the correction pattern.

## Each volume

$$I_{shear}(x,y',z') = \sum_{i=1}^{N_x} \sum_{j=1}^{N_{y'}} \sum_{k=1}^{N_{z'}} \delta(x-i) \cdot \delta(y'-j) \cdot \delta(z'-k) \cdot I_{ijk} \quad (1)$$

is natively recorded in a sheared coordinate system by our microscope and needs to be unsheared for convenient analysis. Here $\delta(x-i)$ is the Dirac delta function and $I_{ijk}$ are the measured image intensity values. During this process, we simultaneously execute two other steps: (1) We axially bin our volumes along z = z' since the effective pixel pitch of the camera (dz'=0.73 µm) is smaller than the axial resolution. (2) We upsample the volume along the y-direction to 1 µm from the native sheared Y'-direction, which is originally sampled at 3.5 µm. This is necessary and beneficial since the projection of the microscope PSF onto the sheared Y', a mixture of axial and lateral resolution, is larger than onto the Y. During unshearing we can therefore recover intermediate Y planes through interpolation.

This is done by computing the unsheared upsampled and axially binned volume at a new coordinate grid with voxel size $0.75 \times 1.0 \times 4.5\,\mu m^3$ (XYZ) by using the following interpolation kernel.

$$I(x,y,z) = \frac{1}{2}\int I_{shear}(x,y',z')tri\left(\frac{z'-z}{s}\right)\left[\left(tri(y-y'-sz')\right. \right.$$
$$\left.\left. + tri(y-y'+1-sz')\right)\right]dy\,dz \qquad (2)$$

Here, $tri(x) = \max(0,1-|x|)$ and s is the slope of the oblique plane. Lastly, in the case of neuro-imaging data, we deconvolve the data using 10 iterations of Richardson-Lucy deconvolution. The kernel was empirically estimated from an average of previously imaged fluorescent beads. The number of iterations of RL deconvolution depends on signal to noise ratio and was manually set after visual inspection.

### Neuroimaging

*Danionella cerebrum* (DC) were kept in commercial zebrafish aquaria (Tecniplast) with the following water parameters: pH 7.3, conductivity 350 µS/cm, temperature 27 °C. We used adult male DC, expressing an histone-tagged (nuclear-localized)[73] GCaMP6s pan-neuronally (HuC:H2B-GCaMP6s x tyr -/-), created by Tol2-mediated transgenesis as described in ref. 14. Fish were placed on a pre-formed agarose mold, which allowed the gill covers to move freely, and immobilized with 2% low melting point agarose. A flow of fresh, aerated aquarium water was delivered to their mouth. They were allowed to recover from anesthesia for 15 min prior to experiment onset. After experiment onset the intensity of the excitation beam was gradually increased over 2 min up to a final power of ≈5.3 mW after the imaging objective to allow for slow habituation. The beam was scanned through one plane in 3 ms and coincided with the 3 ms sensor exposures. Recording and read-out of one plane took 3.1 ms. We could therefore image 332 planes spanning 827.5 µm at 1 Hz volume rate. After post-processing and un-shearing (which involves upsampling along Y and binning along Z) the datasets consisted of volumes with a size of 3024 × 960 × 144 px (XYZ).

### Image registration

In order to analyze time-lapse recordings of whole-brain imaging datasets all volumes had to be motion corrected. One volume from the recording was selected as the template, on which all other volumes were registered. Before estimation all volumes were band-pass filtered with a difference of Gaussian filter ($\sigma_0 = 2$ px, $\sigma_1 = 5$ px) to enhance high frequency features and exclude residual grid artefacts introduced by the face plate. The registration consisted of the estimation of an affine and a non-rigid transformation from estimates of locally rigid displacements. Practically, each volume was chunked into non-overlapping blocks and the three dimensional rigid displacement for each block was then determined via cross-correlation.

In the first iteration, a global affine transformation was fitted onto the obtained coarse displacement field with a blocksize [604 px, 192 px, 28 px] (XYZ). After correction of the coarse affine warp, the volume was again divided into smaller blocks of [32 px, 32 px, 16 px] (XYZ) to estimate a better resolved displacement field. From this field, a full non-rigid displacement field was then obtained via interpolation. Finally, the compound transformation consisting of affine and non-rigid transformation was applied onto the original volume using linear interpolation. We identified time points at which registration failed despite motion correction by thresholding the derivative of the registration metric (global Pearson correlation) at 1.5 standard deviations. These time points were excluded from subsequent analyses.

### Segmentation and calculation of ΔF/F

To identify cell nuclei we used local maxima detection of the local correlation[12,13]. Specifically, we computed the local correlation of voxels with their 26-connectivity surroundings. The resulting map was convolved with a 3D difference of Gaussians kernel ($\sigma_0 = 1$ px, $\sigma_1 = 4$ px). We then detected the local maxima within spherical neighborhoods (⌀ 5 µm), that roughly corresponded to the size of the cell nuclei. The detected maxima were then globally thresholded based on visual inspection. The temporal trace of the remaining points was extracted by calculating the mean of a Gaussian footprint (FWHM = 5 µm) centered at every point. For every trace a baseline was computed by consecutively filtering with a median (7 s), minimum (101 s), and a Gaussian filter ($\sigma$ = 101 s). The ΔF/F was then computed with respect to this baseline and centered. To reject noise traces we used a spectral SNR criterion based on the fact that shot noise is broad-band and signals tend to have more power in lower frequencies (we calculated the ratio between average power at <0.25 Hz and average power at ≥ 0.25 Hz; we set the inclusion threshold to ≥ 1.3). To filter out artifacts from cells near the edge of the FOV that are not visible in all frames due to the global motion of the fish, we removed traces with dF/F values outside the range −100%…2000%. This data was used for all subsequent analyses.

When showing traces in a 2D plot (also called "Rastermaps"[74]), we used 1D Ward clustering to arrange traces by correlation.

### Spatial scale of correlations

For each recording of n cells, we calculated the n×n matrix of pairwise correlations, the n×n matrix of pairwise distances between their coordinates (each matrix containing data from n·(n−1)/2 unique non-identical pairs) and determined the average correlation value for each distance bin.

To create the local neighborhood correlation maps, we calculated the pairwise correlation between a cell and the average activity within a 50 µm distance from this cell−excluding cells closer than 25 µm, a conservative threshold to avoid possible influence from optical cross-talk).

### Prediction of brain-wide activity

All predictions of cellular activity were based on cross-validated multivariate regression. To cross-validate, we split the data along the time dimension into a training dataset (60% of time points), a held-out validation dataset (20% of time points) and test set (20% of time points). For every block of 5 minutes, we assigned the first 3 minutes to training data, followed by 1 min validation data and 1 min test data. We further split data along the cell dimension, randomly selecting 10% of all cells as target cells, and using the remaining 90% of cells to create predictors. This analysis was repeated for each fish.

In principal component regression, predictors consisted of principal components of the training dataset. To create predictor voxels, we discretized the brain into non-overlapping cubic bins, averaged all cellular training dataset activity assigned to a bin, and considered all non-empty bins as potential predictors voxels. For Fig. 4b, c, we considered all non-empty predictor voxels and only varied the spatial scale of discretization. For Fig. 4d we varied both voxel size and predictor number, by random sampling among all potential predictor voxels.

With the exception of principal component regression, we used linear Ridge regression, predicting target cell activity as $Y_{predicted} = X_{test/validation}(X_{train}^T X_{train} + \alpha I)^{-1}X_{train}^T Y_{train}$ (X and Y are matrices of shape timepoints × predictors and timepoints × targets, I is the identity matrix) via double cross-validation.

We determined the regularization parameter α via cross-validation between training and validation dataset and used this parameter in a second cross-validation step between training and test dataset. In case in which we did not de-couple the voxel size from the predictor number we scaled each predictor dataset by a global factor to have the same average standard deviation as the raw dataset to avoid setting individual α parameters for different predictors. The variance

explained per target cell was calculated as $R^2 = 1 - \frac{SS_{res}}{SS_{tot}}$, where $SS_{res}$ corresponds to the sum of squares of the residuals ($y_{predicted} - y_{test/validation}$) and $SS_{tot}$ corresponds to the total sum of squares of the held out data.

Random selection of 10% target cells (without replacement), and prediction of their activity was repeated until all cells were covered. This process was in turn repeated 10 times with new random seeds, resulting in 100 batches. Brain-wide explained variance was calculated as the variance-weighted average of cellular $R^2$ values, averaged across all batches.

### Calculation of distortion and similarity measures

The dF/F traces of each recording constitute a $t \times n$ matrix $X$. We followed the general rationale of computing a $t \times t$ dimensional pairwise distance matrix $D$ between the t timepoints by randomly selecting 50% of the population of cells. We then compared this matrix with a second distance matrix $D_{red}$ computed from a subset of voxelised traces belonging to the held-out 50% of the population.

For the distortion measure we computed matrices $D$ and $D_{red}$ by computing the pairwise Euclidean distance of all t timepoints in $n/2$-dimensional space. The distortion stress was then calculated as $\sum (D - kD_{red})^2 / \sum D^2$. Here $k$ was the least-squares estimate of the global scalar parameter obtained for each embedding.

In case of the representational similarity we computed $D$ and $D_{red}$ as the Pearson correlation between the population vectors of all t timepoints in $n/2$-dimensional space. We then computed the similarity as the Pearson correlation between $D$ and $D_{red}$.

### Statistics & reproducibility

We recorded from a total of ~150k neurons across 6 animals. The number of cells corresponds to the total number of cells detected (see Methods). The sample size was chosen based on the standards in the field. Time points in which motion could not be corrected were excluded from the analysis. We reproduced our findings across all experimental animals. Randomization was used to create a control dataset in Fig. 4. There was no group allocation which needs to be randomized. There was no group allocation which would have required blinding.

### Reporting summary

Further information on research design is available in the Nature Portfolio Reporting Summary linked to this article.

## Data availability

Processed calcium imaging data (activity traces and coordinates of segmented cells) have been deposited in the G-Node at: https://gin.g-node.org/danionella/hoffmann_et_al_2023 Source data are provided with this paper.

## Code availability

Code used for data analysis is available on GitHub at: https://github.com/danionella/hoffmann_et_al_2023.

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

## Acknowledgements

We thank John-Dylan Haynes, Susanne Schreiber, Tatiana Engel, and Alexandre Mathy for the discussions as well as Caroline Berlage, Thomas Chaigne, Verity Cook, Fabian Voigt, Andrew Plested, Michael Brecht, Isaac Kauvar, Marius Pachitariu, Peter Rupprecht, and Ruben Portugues for comments on the manuscript. Data were analyzed on

the Berlin Institute of Health high-performance compute cluster. B.J. acknowledges support by the German Research Foundation (DFG, projects EXC-2049-390688087 and 432195732) the European Research Council (ERC2016-StG-714560, ERC2021-CoG-101043615), the Einstein Foundation (EPP-2017-413) and the Alfried Krupp Foundation.

## Author contributions

Conceptualization: M.H., B.J., Investigation: M.H., J.H., J.V., L.R.; Formal analysis: M.H., B.J.; Writing—original draft: M.H., B.J.; Writing—review & editing: M.H., J.H., J.V., B.J.; Supervision: B.J.

## Funding

## Competing interests

The authors declare no competing interests.
