## [Peer Review File · Nature Communications]

Reviewers' Comments:

Reviewer #1:

Remarks to the Author:

I support publication with no further modifications.

I am strongly enthusiastic about the optical engineering. With that alone I would support publication.

I am enthusiastic about the neuroscience component. The authors are probably more enthusiastic about it than I am, but that's okay (and I may very well be off-base). (1) It is certainly added value as a demonstration, and (2) the revised presentation of that part is clearer to me now. I have a better idea of what the authors are saying. It is well done.

This is a very nice manuscript, and I hope to see it in print soon. I thank the authors for the revisions, the response, and congratulate them on a job well done.

Reviewer #3:

Remarks to the Author:

The authors have addressed my concerns about the optical implementation and their statistical analysis. The revised manuscript makes this clearer and appears fair to me regarding other microscope technologies. I also appreciate the detailed answers to concerns and suggestions on data analysis. There were many, and each was addressed with considerable care.

Importantly, the analytical part of the paper is much improved now - the authors have done alternative analysis and are much clearer on their methodology in the text.

As such, it is an inspiring and thought-provoking article.

Minor comments: Can the authors estimate the excitation NA for their light-sheet? With an NA 0.8 objective, the axial resolution may in part stem from the detection PSF? This is more a technical detail, which could be in favor of the author's approach (I would assume that in the work of Wejun Shao et al, *Optica* 2022, the light-sheet NA is more limiting on the axial resolution).

Other minor point: to the naïve reader, it might at first glance appear that the limiting half angle for traditional OPM objectives is 45 degrees, but it appears to be 30 degrees. I did not readily see this from Figure 1 but had to make a more extended cartoon with some additional angles. This could be added in the supplementary. But this is only a suggestion.

Sincerely,
Reto Fiolka

Reviewer #5:

Remarks to the Author:

In general, the revised manuscript is greatly improved. I am convinced by most of the responses to my comments. I also appreciate the additional data and figures, including the Sup Table 1. My only remaining concern relates to point #1 of my previous report. As the authors strongly emphasize, their results and conclusions could not be reached without a brain-wide analysis at cellular resolution. It is then critical to thoroughly demonstrate that blazed OPM provides sufficient

spatial resolution and imaging depth to segment all cells in the entire *Danionella* brain. Before publication, the authors should address the following points:

(i) First, I am now confused about the actual size of the *Danionella* brain, the achievable imaging volume at cellular resolution and the imaging depth at which cells can be segmented using blazed OPM. Indeed, in the first version of the paper, the authors stated on p4 that the typical size of the adult brain is $2.5 \times 1 \times 0.8 \text{ mm}^3$ corresponding to 2 mm^3 of total volume and a maximum imaging depth of $800 \mu\text{m}$. In the revised version, this is replaced by "we need to sample a volume of $2 \times 1 \times 0.5 \text{ mm}^3$ " (p2), corresponding to twice smaller total volume (1 mm^3) and only $500 \mu\text{m}$ max depth. Is this a mistake or does it mean only a sub-volume of the brain has been recorded? In the latter, it compromises all claims about whole brain imaging and analysis, which is central to this study. In Fig 1, the scale bar suggests the brain is significantly larger than $500 \mu\text{m}$ in depth, closer to $800 \mu\text{m}$. If $800 \mu\text{m}$ is indeed the thickness of the brain, the authors should provide images and cellular resolution demonstration as this depth in Supp. Fig. 4-6 (not limited to $\sim 500 \mu\text{m}$).

(ii) I am not convinced Supp. Fig. 5 supports that cells can be segmented throughout the brain, especially the deepest ones. First, segmentation results should be displayed, not only raw or local covariance images. From the quality of presented data, it is difficult to believe all cells can be segmented. A quantitative approach would be required, for instance by comparing segmentation using blazed OPM and a high-resolution / high-imaging depth microscopy (with smaller FOV, but providing a local ground truth). Finally, since the axial resolution is specifically degraded in blazed OPM, the reconstructed images and segmentation results should be shown in the axial direction (orthogonally to the plane shown in Supp. Fig 5) to be convincing, especially in dense cell area. Are deep cells ($800 \mu\text{m}$) still resolved and well segmented in the axial direction?

(iii) Supp. Fig. 6 demonstrates the axial resolution is significantly degraded at $400 \mu\text{m}$ depth, which should be even worse at $800 \mu\text{m}$ (to show/quantify). It is surprising that such axial resolution is sufficient to segment all cells. If isolated cell nuclei can be identified in depth and displayed, what about dense cell areas?

(iv) Supp. Fig 5-6 are not commented in the main text. For instance, it should be clearly mentioned the measured spatial resolution using beads in a gel is degraded deep in the brain as shown in these supp. Figures.

(v) Finally, I still believe that the sentence "[we] achieve imaging volumes up to 4 mm^3 " in the discussion (p11) is misleading since the paper is about brain imaging and beads in a gel do not mimic brain tissue optical properties. If the authors cannot demonstrate 4 mm^3 imaging in a larger brain than that of *Danionella*, they should at least provide here the actual figure. For instance, they could indicate it in the next sentence of the discussion: "These advantages enabled the first 1 mm^3 volumetric recording of brain-wide neuronal activity in an adult vertebrate." (or 2 mm^3 , see point (i) above)

Reviewer #1:

I support publication with no further modifications.

I am strongly enthusiastic about the optical engineering. With that alone I would support publication.

I am enthusiastic about the neuroscience component. The authors are probably more enthusiastic about it than I am, but that's okay (and I may very well be off-base). (1) It is certainly added value as a demonstration, and (2) the revised presentation of that part is clearer to me now. I have a better idea of what the authors are saying. It is well done.

This is a very nice manuscript, and I hope to see it in print soon. I thank the authors for the revisions, the response, and congratulate them on a job well done.

Thank you, also for your constructive critique in the last round, which helped us improve the quality of the manuscript.

Reviewer #3:

The authors have addressed my concerns about the optical implementation and their statistical analysis. The revised manuscript makes this clearer and appears fair to me regarding other microscope technologies. I also appreciate the detailed answers to concerns and suggestions on data analysis. There were many, and each was addressed with considerable care. Importantly, the analytical part of the paper is much improved now - the authors have done alternative analysis and are much clearer on their methodology in the text.

As such, it is an inspiring and thought-provoking article.

Thank you. We appreciate your suggestions and are glad to hear that they have been addressed.

Minor comments: Can the authors estimate the excitation NA for their light-sheet? With an NA 0.8 objective, the axial resolution may in part stem from the detection PSF? This is more a technical detail, which could be in favor of the author's approach (I would assume that in the work of Wejun Shao et al, Optica 2022, the light-sheet NA is more limiting on the axial resolution).

The excitation NA is ~ 0.03 , with a beam waist of $\sim 7.3 \mu\text{m}$ at an angle of $\varphi_{16x}=33^\circ$, corresponding to an axial profile of $\sim 13.4 \mu\text{m}$, which is why we believe the axial resolution is dominated by the excitation PSF. We added the detection NA to the methods section (next to the beam waist information).

Other minor point: to the naïve reader, it might at first glance appear that the limiting half angle for traditional OPM objectives is 45 degrees, but it appears to be 30 degrees. I did not readily see this from Figure 1 but had to make a more extended cartoon with some additional angles. This could be added in the supplementary. But this is only a suggestion.

Thank you for this suggestion. We added such a drawing to Supplementary Figure 1.

Sincerely,
Reto Fiolka

Reviewer #5:

In general, the revised manuscript is greatly improved. I am convinced by most of the responses to my comments. I also appreciate the additional data and figures, including the Sup Table 1.

Thank you for taking the time to consider our revisions and answers.

My only remaining concern relates to point #1 of my previous report. As the authors strongly emphasize, their results and conclusions could not be reached without a brain-wide analysis at cellular resolution. It is then critical to thoroughly demonstrate that blazed OPM provides sufficient spatial resolution and imaging depth to segment all cells in the entire *Danio rerio* brain.

It is important to distinguish between “brain-wide” and “all cells”. In the context of neuronal recordings, the term “brain-wide” is commonly used to refer to cells sampled across many regions across the brain – it does not imply sampling every cell (for a prominent recent example, where brain-wide data has been recorded much more sparsely, and sequentially, see: doi.org/10.1101/2023.07.04.547681). This is different from the use of “whole-brain”, a term which we consciously excluded from our manuscript.

We indeed do not record from all cells and we do not claim to do so. Our manuscript states the number of segmented active cells per animal and compares it to the literature value for the estimate of total cells in the brain, so we hope that the sampling statistics are communicated transparently (we mention “17 k - 41 k spontaneously active neurons per animal, of an estimated total 650 k neurons [Schulze 2017]”).

Importantly, our conclusions do not require sampling from every single cell in the brain. To illustrate this, let's consider the analysis of figure 4 and the effect of an increased number of predictor or target cells. First, one could rewrite our analysis of Figure 4 to calculate the R^2 for each target cell sequentially and then average (this would be inefficient computationally, but conceptually identical). Thus, increasing the number of target cells would reduce the noise of the average R^2 , but should not change the result. Second, recording more cells would allow us to increase the number of predictor cells, i.e. extend the range of the x-axis in figure 4d. This would be welcome, but the range recorded by us (over 5% of all the cells in the brain, orders of magnitude higher than the fraction of cells recorded in typical neuropixel experiments) was already sufficient to observe the constancy of the R^2 , which we report.

*While our conclusions don't rely on sampling *all* cells, they do rely on sampling simultaneously from many brain regions (“brain-wide”) and sampling their location. The reviewer is right that we lose resolution towards the bottom of the brain (as shown on Figures S5, S6, and the new Figure S7) and therefore undersample there. This is an unfortunate practical limitation shared by larval zebrafish literature. Our updated manuscript now states this more clearly and explicitly in the main text and in the discussion.*

Results:

[...] we recorded spontaneous activity from up to 41k active neurons throughout distant brain regions up to a depth of ~400-500 μ m, including the brainstem, tectum, pallium and diencephalon, excluding the ventral hypothalamus (Figure 1 f-i , Supplementary Figures 3-7 and Supplementary Video 1).”

Discussion:

“As with other linear microscopy techniques, the depth penetration of OPM is limited by optical scattering, which prevents us from recording cells in dense nuclei and ventral most areas of the brain below 400-500 μm , such as the ventral hypothalamus (see Supplementary Figures 4-7).”

Before publication, the authors should address the following points:

(i) First, I am now confused about the actual size of the *Danio* brain, the achievable imaging volume at cellular resolution and the imaging depth at which cells can be segmented using blazed OPM. Indeed, in the first version of the paper, the authors stated on p4 that the typical size of the adult brain is $2.5 \times 1 \times 0.8 \text{ mm}^3$ corresponding to 2 mm^3 of total volume and a maximum imaging depth of $800 \mu\text{m}$. In the revised version, this is replaced by “we need to sample a volume of $2 \times 1 \times 0.5 \text{ mm}^3$ ” (p2), corresponding to twice smaller total volume (1 mm^3) and only $500 \mu\text{m}$ max depth. Is this a mistake or does it mean only a sub-volume of the brain has been recorded? In the latter, it compromises all claims about whole brain imaging and analysis, which is central to this study. In Fig 1, the scale bar suggests the brain is significantly larger than $500 \mu\text{m}$ in depth, closer to $800 \mu\text{m}$. If $800 \mu\text{m}$ is indeed the thickness of the brain, the authors should provide images and cellular resolution demonstration as this depth in Supp. Fig. 4-6 (not limited to $\sim 500 \mu\text{m}$).

*The reviewer is right to be confused, as measures of brain size weren't communicated well, nor defined sufficiently. We have an ongoing project on brain anatomy, which allowed us to define and determine these numbers with more confidence. Along the lateral dimension, the brain size is easiest to define, as it is the width of the optic tectum ($\sim 1.2 \text{ mm}$ from tip to tip). Along the anteroposterior axis, the anterior end of the forebrain is well defined, but the transition between hindbrain and spinal cord lacks an exact demarcation. In response to the reviewer's comment, we reviewed existing zebrafish anatomical maps with a focus on the hindbrain/spinal cord boundary, and concluded that the average length of the *Danio* brain is approximately $\sim 2.2 \text{ mm}$, a number which we now use throughout the manuscript. The maximum dorsoventral extent is $\sim 0.65 \text{ mm}$, from the dorsal end of the midbrain to the ventralmost nucleus of the hypothalamus. Since the ventralmost areas of the hypothalamus are tapered towards a ventro-medial tip, and constitute a small fraction of the total brain volume, they are often excluded from analyses in zebrafish (see e.g. Marques et al. Nature 2020), but the reviewer is right to complain: we should stick to a definition and report accordingly. We now report the full depth of the brain (and also mention explicitly that we do not resolve cells in the ventral hypothalamus, see response above).*

Results:

*“In order to record cellular activity with calcium sensors throughout the brain of adult *Danio* we need to sample a volume of approximately $2.2 \times 1.2 \times 0.65 \text{ mm}^3$ at $\geq 1 \text{ Hz}$.”*

Finally, we thank the reviewer for alerting us to the incorrect scale bar in Figure 2, an error that must have occurred during figure resizing. This is now corrected, double-checked and consistent with the other statements and figures throughout the paper.

(ii) I am not convinced Supp. Fig. 5 supports that cells can be segmented throughout the brain, especially the deepest ones.

This is correct. As we explain above, we do not claim or state that we segment all cells. The purpose of Figures S5 and S6 is to show imaging quality at various locations and the quality indeed gets worse with depth, as one would expect for a 1P imaging method. Our previous version listed brain areas in which cells can be resolved. This description has been extended to explicitly mention the imaging depth limit of approximately 400-500 μm and that we cannot resolve cells in the ventral hypothalamus (see quoted text above).

First, segmentation results should be displayed, not only raw or local covariance images.

In addition to the segmentation results displayed in the new Figure S7 (described below), we also included a new supplementary video showing a 3D stack of raw fluorescence, local covariance (the basis for segmentation) and markers for segmented cells.

From the quality of presented data, it is difficult to believe all cells can be segmented. A quantitative approach would be required, for instance by comparing segmentation using blazed OPM and a high-resolution / high-imaging depth microscopy (with smaller FOV, but providing a local ground truth).

We do not claim to segment all cells (see above), but we agree that a comparison with 2P microscopy (2PM) would be helpful. To address the reviewer's suggestion, we performed sequential OPM and 2PM recordings. As our segmentation is activity-based, we performed these measurements in vivo, under similar conditions as in the rest of the paper (spontaneous activity). The main difference was the recording time, which was lowered to 10 min due to the limitations of 2PM: because 2PM is much slower than OPM, we recorded 2P data sequentially from a handful of planes distributed along the z-axis. To keep the total experimental time $\leq 1\text{h}$, we recorded OPM data and then several 2P planes for 10 mins each. Both OPM volumes and 2P planes were acquired at the same rate of 1 Hz. We then identified the corresponding 2P planes in the OPM volume and performed activity-based segmentation on these planes for both modalities. The locations of the planes, the number of cells detected at each depth and the quality of the local covariance maps used for segmentation are shown in the new Figure S7. After correcting for the larger axial extent of OPM compared to 2PM (which would give OPM an unfair advantage in terms of cells detected per plane), we estimate that OPM detected on average $\sim 40\%$ of the cells that were detected with 2PM. Apart from the difference in resolution, another explanation is the higher SNR of 2PM (which can also be seen in the different levels of noise of the local covariance maps): while the 2PM data was acquired at 1 plane per second, OPM scanned one entire 3D volume per second. The new figure also shows a decline of the cell count after $\sim 400\text{-}500\ \mu\text{m}$. We take this value ($400\text{-}500\ \mu\text{m}$) as the approximate depth limit, mentioned in the updated results and in the discussion.

Finally, since the axial resolution is specifically degraded in blazed OPM, the reconstructed images and segmentation results should be shown in the axial direction (orthogonally to the plane shown in Supp. Fig 5) to be convincing, especially in dense cell area. Are deep cells (800 μm) still resolved and well segmented in the axial direction?

(iii) Supp. Fig. 6 demonstrates the axial resolution is significantly degraded at 400 μm depth, which should be even worse at 800 μm (to show/quantify). It is surprising that such axial resolution is sufficient to segment all cells. If isolated cell nuclei can be identified in depth and displayed, what about dense cell areas?

The data throughout Supp. Fig. 4-6 (and the new Figure S7) are representative of the volume that we image. We added side views to the images in Supplementary figure 5, which include areas of varying density and illustrate that resolution gets worse at depth. We also included an additional supplementary video showing activity-based segmentation results (and the quality of the local correlation, which is the basis for segmentation) across the entire brain, including areas of varying density. As mentioned above, we don't claim to segment all cells and we communicate the number of cells recorded. We now also mention the approximate imaging depth limit of 400-500 μm (and the dorsoventral extent of the brain around 650 μm).

(iv) Supp. Fig 5-6 are not commented in the main text. For instance, it should be clearly mentioned the measured spatial resolution using beads in a gel is degraded deep in the brain as shown in these supp. Figures.

We now explicitly refer to these figures in the Results and in the Discussion (where we mention the degradation of resolution due to scattering – see text quoted above).

(v) Finally, I still believe that the sentence “[we] achieve imaging volumes up to 4 mm^3 ” in the discussion (p11) is misleading since the paper is about brain imaging and beads in a gel do not mimic brain tissue optical properties. If the authors cannot demonstrate 4 mm^3 imaging in a larger brain than that of *Danio rerio*, they should at least provide here the actual figure. For instance, they could indicate it in the next sentence of the discussion: “These advantages enabled the first 1 mm^3 volumetric recording of brain-wide neuronal activity in an adult vertebrate.” (or 2 mm^3 , see point (i) above)

The optical performance of blazed OPM in clear media (bead stack) versus turbid media (tissue) is indeed different. However, we still think that reporting the FOV in a clear medium is important for two reasons: On one hand our paper is also about a contribution to the OPM field and in this paragraph we compare our image volume to Refs 23,26,27 and 29, all of which report their maximal FOV based on the imaging of cleared or thin media. On the other hand other scientists may wish to adapt this imaging technique to their needs (possibly outside of the area of brain imaging) and therefore might want to know the achievable volume under best conditions. The reviewer is correct, however, that especially in light of the second reason we should more explicitly state that the performance of this one photon technique is degraded as we image in turbid media. As mentioned above, we now explicitly state this in the Discussion. We also clarified that our report of a FOV of up to 4 mm^3 refers to clear media.

Reviewers' Comments:

Reviewer #3:

Remarks to the Author:

The authors have addressed my remaining (voluntary) points.

I recommend publication as is.

Sincerely,
Reto Fiolka

Reviewer #5:

Remarks to the Author:

I thank the authors for their thorough and well-documented reply regarding the imaging depth limitations of blazed OPM. I really appreciate the additional data, figures and movie. They clearly addresses my last concern. I am now ready to support publication without further revision.